# AI-Driven Talent Matching Predicting Employee Retention Through Candidate Attribute Analysis

Yasir Imtiaz Shami  and Tosin Adewumi

Luleå University of Technology, Sweden.
{yassha-4@student.ltu.se, tosin.adewumi@ltu.se}

## Abstract

This study presents an AI-driven talent matching framework that combines semantic similarity with retention-aware modeling. The system uses transformer-based embeddings to match candidates and jobs, while also predicting retention through psychometric traits, employment history, and education. The results show that semantic matching alone provides strong rankings, but adding retention scoring improves shortlist quality.

## 1  Introduction

Employee retention remains a pressing challenge in human capital management. According to recent industry statistics, nearly 38% of new employees leave within 12 months of being hired [1]. This early turnover incurs financial costs for organisations. Recent advances in AI and NLP have led to more advanced hiring systems that move beyond simple filtering. These AI-driven approaches include semantic matching engines that utilise deep learning-based embeddings [2, 3]. Although these techniques improve the quality of hiring in terms of relevance, they still lack the capacity to predict retention, which is a critical indicator of hiring effectiveness. The primary goal of this study is to develop and evaluate an AI-driven Talent Matching Framework that incorporates retention features into candidate job matching. Our **research questions** are (1) Which candidate attributes (e.g., personality traits, prior employment history, skills, education) are significantly associated with talent retention in new job roles and (2) How can retention-predictive candidate data be effectively collected, structured, and integrated into the talent evaluation process? Hence, the objectives are:

- Identify candidate attributes predictive of long-term retention.

- Design and implement an AI-driven Talent Matching Framework that integrates these features into a candidate ranking system.

- Evaluate the framework.

## 2  Methodology

Our framework followed a structured process consisting of data preparation, feature extraction, embedding generation, semantic matching, retention scoring, and unified ranking. This study utilises a primary proprietary dataset provided by a Swedish hiring firm. The dataset includes approximately 7,000 candidate profiles and several thousand job descriptions. It was split in the ratio 80%, 10%, and 10% for training, dev, and test sets, respectively. To compute the degree of similarity between a job and a candidate, the cosine similarity is used, as a metric and given by Equation 1.

$$\text{cosine\_similarity}(\mathbf{A}, \mathbf{B}) = \frac{\mathbf{A} \cdot \mathbf{B}}{\|\mathbf{A}\| \|\mathbf{B}\|} \qquad (1)$$

Where $\mathbf{A}$ and $\mathbf{B}$ are the embedding vectors for a job description and a candidate profile, respectively. Cosine similarity measures the cosine of the angle between the two vectors, providing a value between $-1$ and 1, where 1 indicates perfect semantic alignment and 0 indicates orthogonality (no similarity). This work integrates a separate **retention modeling** layer that estimates the candidate's likelihood of staying, based on historical behavior, educational attainment, and psychometric indicators. Table 1 provides the weights for the psychometric indicators, used in computing a retention likelihood score through a weighted sum. A final composite retention score that averages (1) the psychometric score, (2) tenure score, and (3) degree score, is then calculated, based on the literature [4–6].

**Table 1.** Weights for Psychometric Scoring

| Personality Trait | Weight |
|---|---|
| Conscientiousness | 0.35 |
| Agreeableness | 0.30 |
| Emotional Stability | 0.20 |
| Openness | 0.10 |
| Extraversion | 0.05 |

As additional metric, we used NDCG@K, derived from the Discounted Cumulative Gain (DCG). It evaluates the ranking quality by assigning higher importance to correctly ordering more relevant candidates near the top of the list. DCG is given by

**Table 2.** Case Study: Top Candidates Matched to Job ID 0

| Cand. ID | Sim. Score | Ret. Score | Final Score | Candidate Summary (Truncated) |
|----------|-----------|-----------|-------------|-------------------------------|
| 1307 | 0.6829 | 0.5029 | 0.5749 | Provides counseling/admissions; 400 ECTS from University X; experience as developer with Company X and others. |
| 1274 | 0.6816 | 0.5016 | 0.5736 | Experienced school principal in multicultural areas; emphasizes collaboration and student success. |
| 2069 | 0.7111 | 0.4374 | 0.5469 | HR/Talent Acquisition expert for Nordic market; experienced recruiter with global MNCs. |
| 466 | 0.7039 | 0.4336 | 0.5417 | Teaching + Payroll/HR experience in international companies (EY, Credit Suisse); strong admin skills. |
| 2535 | 0.6927 | 0.2613 | 0.4339 | Global Talent Specialist; built IT/QA and recruitment teams; strong project background. |

Equation 2 and NDCG@K by Equation 3, where $rel_i$ is the ground truth relevance (in our case, the predicted retention score) of the candidate ranked at position $i$. To obtain a normalized score, this DCG is divided by the Ideal DCG (IDCG), which represents the best possible ordering of candidates. This yields a score between 0 and 1, with higher values indicating that highly relevant candidates are placed higher in the ranked list.

$$\text{DCG@}K = \sum_{i=1}^{K} \frac{rel_i}{\log_2(i+1)} \qquad (2)$$

$$\text{NDCG@}K = \frac{\text{DCG@}K}{\text{IDCG@}K} \qquad (3)$$

## 3 Results and Discussion

Table 2 shows example predictions and Table 3 shows the test performance across five different seeds. We can see that NDCG@3 (with retention) is stable with a mean of 0.9996 ($\sigma = 0.0001$), more or less like the baseline. Retention does not hurt performance.

**Table 3.** Test Performance Metrics Across Runs (Top-3 Recommendations)

| Run | NDCG@3 (Retention) | Baseline |
|-----|--------------------|-----------| 
| 1 | 0.9995 (0.0023) | 0.9995 (0.0023) |
| 2 | 0.9995 (0.0020) | 0.9995 (0.0020) |
| 3 | 0.9996 (0.0017) | 0.9996 (0.0017) |
| 4 | 0.9997 (0.0013) | 0.9997 (0.0013) |
| 5 | 0.9997 (0.0011) | 0.9997 (0.0011) |
| **Mean** | **0.9996** | **0.9996** |
| **sd** | 0.0001 | 0.0001 |

As an example, as shown in Table 2, the system selected the top 5 candidates ranked for a particular job. The rankings incorporate both semantic similarity and retention scores. Each candidate is described with their summary and the computed matching metrics.

## 4 Conclusion

This study introduced an AI-driven talent matching framework that combines semantic similarity with retention-aware modeling. The framework integrates psychometric inference from the Five-Factor Personality Theory, employment history, and educational background as core attributes to support predictive hiring. The results showed that semantic embeddings alone provide strong rankings, but adding retention scoring can reshape shortlists in useful ways. Candidates with higher stability potential were promoted, making the rankings more practical for recruitment. Case studies and the interactive dashboard further demonstrated transparency and applicability.

Some limitations remain: inferring personality traits from text can be noisy. Also, incomplete datasets reduce prediction accuracy. Psychometric inference from CVs should not replace dedicated personality tests, but in the absence of such assessments, personality indicators can be inferred from resumes or reference letters. The proposed framework can help reduce early turnover, lower recruitment costs, and support the creation of more stable and resilient teams.

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
