# OpenReview forum: "AI-Driven Talent Matching Predicting Employee Retention Through Candidate Attribute Analysis"
_NLDL.org/2026/Abstracts_Track — NLDL 2026 Abstracts_

### Official Review · Reviewer_zhsJ · 2025-10-27

**Soundness:** 3
**Correctness:** 3
**Rating:** 4
**Confidence:** 4

**Summary:**

This abstract presents an AI-driven framework that integrates semantic similarity models with retention-aware modeling for improved talent matching and employee retention prediction. The system uses transformer-based embeddings for job–candidate matching and includes psychometric, educational, and employment history data to estimate long-term retention likelihood. Experimental results show that while semantic similarity alone provides strong rankings, incorporating retention modeling improves shortlist quality and recruitment outcomes.

**Strengths:**

Novel Integration of Retention Modeling: The paper introduces an innovative framework that bridges semantic similarity in candidate-job matching with retention prediction — a relatively underexplored intersection of NLP and HR analytics.

Methodological Soundness: The authors clearly outline a structured pipeline covering data preparation, embedding generation, similarity computation, and retention scoring. The use of cosine similarity and weighted psychometric features grounded in the Five-Factor Personality Model adds methodological transparency.

Relevance and Practical Impact: By focusing on retention prediction, the work goes beyond relevance ranking and addresses a real organizational pain point — early turnover.

Empirical Evaluation: The inclusion of NDCG@K metrics, stability across multiple random seeds, and a concrete case study table demonstrates careful quantitative validation.

Clarity and Organization: The abstract follows a coherent structure (Introduction, Methodology, Results, Conclusion). Tables and formulas are presented clearly, supporting readability and reproducibility.

Societal and Industrial Significance: The research has potential applications in AI-driven recruitment platforms and organizational HR analytics, promising cost reductions through improved candidate-job fit and retention forecasting.

**Weaknesses:**

Limited Experimental Depth: The evaluation relies heavily on proprietary data from a single company and focuses on ranking stability (NDCG@3) rather than predictive generalization. More diverse datasets or ablation studies would strengthen the results.

Minimal Quantitative Gains: The reported NDCG@3 values (≈0.9996 for both baseline and retention-enhanced models) show negligible differences, which weakens the claim of improved shortlist quality.

Personality Inference Reliability: Inferring psychometric traits from resumes is acknowledged as noisy. Without validation against ground-truth personality assessments, the reliability of retention prediction is questionable.

Ethical Considerations: The paper could more deeply discuss ethical implications, such as bias in retention prediction (e.g., socioeconomic or gender bias) and transparency in automated hiring systems.

Incomplete Statistical Reporting: Standard deviations and confidence intervals are shown but limited discussion is given on statistical significance or error analysis.

Scope Limitation: Since this is an abstract, implementation details and architectural specifics (e.g., embedding model type, dimensionality, or retention classifier structure) are missing, which limits replicability and peer evaluation.

---

### Official Review · Reviewer_uDK2 · 2025-11-01

**Soundness:** 1
**Correctness:** 2
**Rating:** 2
**Confidence:** 4

**Summary:**

The abstract presents a talent matching framework based on AI methods which also includes a retention score for the candidates. This is then used on a proprietary dataset from a Swedish company and evaluated according to multiple metrics. The methods involve calculating the cosine similarity between the embeddings of the job description and candidate profiles and using a psychometric score, a tenure score and a degree score to give a final retention score.

**Strengths:**

The abstracts provide a method for automating hiring and giving a retention score to applicants. If this would be proved to have good performance and low bias, this can be a useful way to increase hiring efficiency. Several metrics are used for evaluation and a proprietary dataset is used, showing that there are some incentives to create the proposed framework.

**Weaknesses:**

The main weakness of the presented work is the soundness of the whole framework. Without the premises clearly stated or argued for, the whole framework loses credibility and usefulness. Please see the bullet points below for greater detail.

- An underlying premise that is not mentioned is that the input data contains predictive information about the response. This would be that the job application and work history would be predictive of job retention. This premise should be clearly stated, but it is also highly questionable. Although work history would be able to predict some sort of retention based on the amount of jobs the candidate has had over a time period, this data should probably not be used directly to predict retention due to not being the direct reason for changing jobs. There are multiple crucial factors influencing how likely a candidate is to change jobs that are not included here. Some of them are how the work environment in the job was, why the candidate changed jobs, the personal and medical conditions of the candidate and the global economic landscape. For instance, assume a highly skilled candidate is applying that has recently changed jobs because of harassment in their previous job environment or because of the medical condition of close family. This could make the model predict low retention, even though they could be a good candidate without a high future probability of retention. In another example, assume that a particular division of the company that is applying has substantially lower quality of work conditions, making the turnover there higher than other departments. The specific traits of the candidates applying to this department could then be predicted to be negative, even though it is not due to the candidates. If for instance this department was a legal department, and therefore had many candidates with backgrounds in law, the resulting system may be negatively biased against candidates with backgrounds in law. In short, the authors should address and argue for the fundamental premises of the framework, as there are many potential issues regarding them.
- The actual ranking method is not sufficiently clear from the abstract. The author refers to existing literature for details in line 071, but since these metrics are at the core of the entire framework, they should be explicitly described in greater detail. In order to make more room for this, the authors could safely shorten the cosine distance metric explanation in line 052 to 061.
- Although the cosine similarity is introduced in moderate detail, it is not clear how it is actually used in the final scores. It seems like a higher cosine score would result in a better score, but this premise should also be stated and argued for. Why would the cosine similarity between some embeddings of the job description and application directly matter? This would create a positive bias for applications written in the same style and language as the job description, but it would not necessarily mean that the content would be suitable. Using a single embedding for a whole application is also a questionable choice, and the reasoning behind it is not addressed.
- The abstract does not describe how the embeddings for the job descriptions and applications are calculated and with which models.
- The weighting in Table 1 is not justified. Why was this exact weighting chosen? Why are some traits more important than others?
- The abstract does not address any ethical concerns around the framework, of which there are many. AI-models are known to have significant bias, especially in the context of hiring. The paper should address how the models may be biased against certain people with certain backgrounds, genders, names, and other traits, and how this can be mitigated.
- The conclusion mentions that using a CV to infer a psychometric score can be noisy, but it should be discussed if this is a sound method at all. If the AI-model directly favors some psychometric scores over others, the applicants could write their applications and resume using a language that would suggest a higher degree of the desired traits. This would result in the score not being predictive of the actual psychometric traits.

---

### Official Review · Reviewer_XjuV · 2025-11-03

**Soundness:** 2
**Correctness:** 3
**Rating:** 4
**Confidence:** 3

**Summary:**

The objective is to accelerate the recruitment process and increase retention after hiring using AI instead of biased recruiters.

**Strengths:**

This is a 2-line abstract; hard to evaluate the strengths.

**Weaknesses:**

Ethical considerations must be taken when conducting this type of work. No mention in this short abstract.

---

### Decision · Program_Chairs · 2025-11-05

**Decision:**

Accept

**Comment:**

The reviewers found the abstract borderline, yet the PCs believe it will be of interest to the community and should have the opportunity be presented.